# Ginsenoside Rh2 Ameliorates Atopic Dermatitis in NC/Nga Mice by Suppressing NF-kappaB-Mediated Thymic Stromal Lymphopoietin Expression and T Helper Type 2 Differentiation

**DOI:** 10.3390/ijms20246111

**Published:** 2019-12-04

**Authors:** Eunsu Ko, Sungjoo Park, Jun Hyoung Lee, Chang-Hao Cui, Jingang Hou, Myung-ho Kim, Sun Chang Kim

**Affiliations:** 1Department of Biological Sciences, Korea Advanced Institute of Science and Technology, Daejeon 34141, Korea; eunsu_93@kaist.ac.kr (E.K.); parksungjoo@kaist.ac.kr (S.P.); junhlee@kaist.ac.kr (J.H.L.); 2Intelligent Synthetic Biology Center, Daejeon 34141, Korea; seldoms@163.com (C.-H.C.); houjingang1225@126.com (J.H.); 3Laboratory of Liver Research, Graduate School of Medical Science and Engineering, KAIST, Daejeon 34141, Korea; check8x8@kaist.ac.kr

**Keywords:** atopic dermatitis, thymic stromal lymphopoietin, NF-κB pathway, Th2 differentiation, ginsenoside, ginseng

## Abstract

Ginsenosides are known to have various highly pharmacological activities, such as anti-cancer and anti-inflammatory effects. However, the search for the most effective ginsenosides against the pathogenesis of atopic dermatitis (AD) and the study of the effects of ginsenosides on specific cytokines involved in AD remain unclear. In this study, ginsenoside Rh2 was shown to exert the most effective anti-inflammatory action on thymic stromal lymphopoietin (TSLP) and interleukin 8 in tumor necrosis factor-alpha and polyinosinic: polycytidylic acid induced normal human keratinocytes by inhibiting proinflammatory cytokines at both protein and transcriptional levels. Concomitantly, Rh2 also efficiently alleviated 2,4-dinitrochlorobenzene-induced AD-like skin symptoms when applied topically, including suppression of immune cell infiltration, cytokine expression, and serum immunoglobulin E levels in NC/Nga mice. In line with the in vitro results, Rh2 inhibited TSLP levels in AD mice via regulation of an underlying mechanism involving the nuclear factor κB pathways. In addition, in regard to immune cells, we showed that Rh2 suppressed not only the expression of TSLP but the differentiation of naïve CD4+ T-cells into T helper type 2 cells and their effector function in vitro. Collectively, our results indicated that Rh2 might be considered as a good therapeutic candidate for the alternative treatment of AD.

## 1. Introduction

Atopic dermatitis (AD) is a prevalent inflammatory disease induced by multi-faceted interplays of genetic, environmental, and immunological factors [1,2]. Specifically, the pathogenesis of AD is mediated by cytokines secreted from keratinocytes and T helper type 2 (Th2)-polarized T-cells, such as thymic stromal lymphopoietin (TSLP), interleukin 8 (IL-8), and IL-4 [3].

Of these cytokines, TSLP is produced mainly by keratinocytes and plays a critical role in the pathogenesis of AD, acting as a master regulator of allergic inflammation by promoting the differentiation of Th cells into Th2 cells either directly or via interactions with dendritic cells [4,5,6,7,8,9]. TSLP-mediated Th2 responses lead to the secretion of cytokines, including IL-4, IL-5, and IL-13, resulting in B-cell activation, which in turn induces an increase in immunoglobulin E (IgE) levels and the number of eosinophils [10,11]. Those responses increase the susceptibility to the manifestation of the atopic march, a developmental progression of AD, followed by allergic rhinitis and asthma [12]. Accordingly, regulation of the expression of *TSLP* could be critical in preventing not only AD pathogenesis but the development of the atopic march mediated by Th2 responses [5].

AD decreases the patient quality of life and, consequently, the demands for treatments have increased. Therefore, the investigation of natural anti-inflammatory compounds has received special attention due to their previously demonstrated safety and efficiency [13,14,15]. Ginsenosides, the major effective components of *Panax ginseng* and valuable natural compounds, have been well reported for their various pharmacological activities. Ginsenosides are triterpene saponins that consist of a dammarane skeleton with a variety of sugar moieties attached to the C-3 and the C-20 positions [16]. The number, the position, and the type of sugar moieties have been known to contribute to diverse pharmacological potentials of ginsenosides, such as anti-cancer, anti-aging, and anti-inflammatory properties [17,18,19]. As previously reported, administration of red ginseng extract was shown to have an ameliorating effect on AD-like skin lesions by suppressing proinflammatory cytokines and chemokines via inhibition of mitogen-activated protein kinase (MAPK) and NF-κB pathway [20,21]. Additionally, ginsenosides Rg3, Rf, and Rh2 have been reported to inhibit passive cutaneous anaphylaxis and contact dermatitis in a mouse model by suppressing the expressions of cyclooxygenase (COX)-2, interleukin (IL)-1β, tumor necrosis factor-α (TNF-α), and interferon-γ (IFN-γ) [22]. Collectively, these studies have shed light on the possibility that ginsenosides could be applied as anti-AD agents. However, the inhibitory effects of ginsenosides on TSLP as well as the identification of the most effective ginsenosides for relieving AD symptoms have not been sufficiently investigated [23].

In this study, we screened for ginsenosides that ameliorate the production of TSLP and IL-8 in normal human keratinocytes (NHKs). We further examined if the identified ginsenoside, Rh2, markedly relieved the 2,4-dinitrochlorobenzene (DNCB)-induced AD-like skin inflammation in NC/Nga mice. We also investigated if the anti-atopic effects of Rh2 result from the blockade of TSLP production via the NF-κB pathway in keratinocytes and Th2 cell differentiation.

## 2. Results

### 2.1. Rh2 Attenuated Inflammatory Cytokines in Stimulated NHKs

To compare the effects of ginsenosides against AD, we screened 17 kinds of ginsenosides presented in Appendix A [compound K (C-K), F1, F2, gypenoside XVII (G17), gypenoside LXXV (G75), protopanaxadiol (PPD), protopanaxatriol (PPT), Rb1, Rb3, Rc, Rd, Re, Rg1, Rg2, Rg3, Rh1, and Rh2] for inhibition of the production of TSLP and IL-8, which plays a role as the hallmark of acute inflammation by inducing neutrophil infiltration into inflammatory sites [24] in stimulated NHKs. To mimic the AD-like inflammatory condition in vitro, a cocktail of proinflammatory agents, TNF- α, and polyinosinic:polycytidylic acid (Poly I:C) was used [25]. As illustrated in Figure 1a, C-K, F2, G75, PPD, PPT, Rg3, and Rh2 significantly inhibited the production of TSLP in response to TNF- α and Poly I:C. Furthermore, C-K, PPD, Rc, and Rh2 markedly decreased the levels of IL-8 compared with stimulated cells. The ginsenoside Rh2 exhibited the most potent inhibitory effects against the production of both TSLP and IL-8 in similar levels to dexamethasone (DEX), which is widely used in the treatment of AD [26] (Figure 1A–C). Rh2 was thus chosen as the candidate for subsequent experiments. To identify any cytotoxic effects of Rh2, cell viability assays were performed. As shown in Figure 1C, Rh2 in concentrations of up to 10 μM had no cytotoxic effects on the NHKs. Rh2 attenuated the production of TSLP and IL-8 in a dose-dependent manner at concentrations of up to 5 μM, whereas no significant differences in the inhibitory effects of Rh2 were observed from 5 to 10 μM (Figure 1D–E), suggesting that 5 μM of Rh2 was enough to downregulate the inflammatory cytokines. This effect was further confirmed by qRT-PCR analysis, in which resulting administration of 5 μM of Rh2 significantly downregulated the mRNA expression level of *TSLP* (Figure 1F). In addition, the mRNA expression level of *IL-8* was also suppressed by treatment of Rh2 (Figure 1G). To further demonstrate the anti-inflammatory effects of Rh2 on the different inflammatory cytokines, the effects of Rh2 were evaluated using a cytokine proteome profile array. This demonstrated that Rh2 decreased the levels of the C-C motif chemokine ligand 5 (CCL5), granulocyte-macrophage colony-stimulating factor (GM-CSF), intercellular adhesion molecule 1 (ICAM-1), and IL-6, all of which act as proinflammatory cytokines that participate in diverse immune responses (Figure 1H).

### 2.2. Rh2 Safely Ameliorated DNCB-Induced AD-Like Skin Inflammation

To further investigate whether Rh2 could ameliorate DNCB-induced AD-like skin inflammations in NC/Nga mice, AD-like skin lesions were induced by applying DNCB on the dorsal skin of the mice with or without Rh2 and DEX treatments (Figure 2A). Body weight changes of the mice were analyzed to assess whether Rh2 had harmful effects on their health status or induced growth retardation. There were no statistical differences between the changes in the body weight of the normal control and the Rh2-treated group, suggesting the absence of major side effects and the safety of Rh2 for in vivo applications. By contrast, the body weights of the mice in the vehicle and the DEX-treated groups were significantly decreased compared with the normal group (Figure 2B). These results were consistent with previous reports that long term use of dexamethasone (DEX) induces growth retardation. Representative images shown in Figure 2C revealed that the topical application of Rh2 significantly suppressed AD-like skin lesions compared to the vehicle control. We further attempted to evaluate the clinical symptoms of the mice macroscopically by applying the clinical atopic dermatitis severity scores. Accordingly, the AD index of the vehicle control reached 8.9 points, whereas treatment with Rh2 significantly decreased it to 3.3 points (Figure 2C). 

Elevated levels of serum IgE are one of the main characteristics and a critical indicator of AD. DNCB-induced mice showed significantly increased IgE serum levels (157.4 ± 42.4 ng/mL). However, the IgE serum levels were markedly alleviated following treatment with Rh2 (27.2 ± 8.9 ng/mL) (Figure 2D). To determine whether topical applications of Rh2 could influence systemic immune responses, the weight of the spleen was measured because dysregulation of immune response in AD results in the development of splenomegaly owing to the fact that the spleen contains various immune cells and plays a crucial role in modulating immune responses [27]. As shown in Figure 2E, treatment with Rh2 resulted in a significantly reduced spleen weight (3.82 ± 0.25) compared with DNCB-induced enlargement of the spleen (5.02 ± 0.19).

### 2.3. Rh2 Suppressed Epidermis Thickening and Mast Cell and Eosinophil Infiltration

Atopic dermatitis-induced skin lesions have typical microscopic characteristics, including hyperkeratosis, acanthosis, parakeratosis, and infiltration of the inflammatory cells. DNCB-treated mice showed clinical manifestations of AD, such as destruction of the skin tissues and increased epidermal thickness, associated with an increased number of infiltration mast cells and eosinophils. In contrast, treatment of the AD-induced lesions with Rh2 remarkably alleviated these symptoms (Figure 3). These results suggested a potent and regulatory effect of Rh2 not only on keratinocytes but on the overall immune response in AD.

### 2.4. Rh2 Inhibited the Expression of TSLP in Keratinocytes of an AD Mouse Model

We aimed to test whether Rh2 alleviated TSLP in an AD mouse model, in line with Figure 1. Topical applications of DNCB on the skin tissues induced the production of TSLP in the mouse epidermis, whereas administration of Rh2 significantly inhibited it (Figure 4A). To further verify this finding, the inhibitory effects of Rh2 on TSLP in vivo were also confirmed at mRNA and protein expression levels of the control and the treated mouse skin tissues. The mRNA expression of *TSLP* was significantly downregulated by Rh2 treatment in the mouse skin tissues (Figure 4B). In accordance with Figure 4A–B, protein expression of TSLP was also significantly decreased in the mouse skin tissues subjected to topical applications of Rh2 (Figure 4C).

### 2.5. Rh2 Blocked the NF-κB Pathway in Stimulated NHKs

To elucidate the mechanisms of Rh2 responsible for the regulation of the production of TSLP in NHKs, the suppressive effects of Rh2 on the NF-κB pathway, which is known as a major pathway for inducing the production of TSLP [28,29,30], were investigated. Upon induction of TNF-α and Poly I:C, secretion of TSLP from the NHKs was inhibited following treatment with Bay 11-7082, NF-κB inhibitors (Appendix A). This suggested that the activation of NF-κB could be responsible for the regulation of the expression of TSLP. As illustrated in Figure 5A, treatment of NHKs with TNF-α and Poly I:C induced the nuclear translocation of p65. By contrast, the nuclear translocation of p65 was partially blocked by treatment with DEX and Rh2, respectively. Consistent with the results in Figure 5a, Rh2 significantly decreased the NF-κB-stimulated luciferase activity (Figure 5B). Furthermore, Rh2 reduced the phosphorylation of p65 and IκBα significantly, resulting in the inhibition of the degradation of IκBα (Figure 5C–D). These results suggested that Rh2 may inhibit the production of TSLP via the inhibition of the NF-κB signaling pathway in NHKs.

### 2.6. Rh2 Inhibited Th2 Differentiation and Effector Function

TSLP is widely known to induce Th2 responses by elevating Th2 cytokines such as IL-4, IL-5, and IL-13 in allergic diseases, including AD [31]. We further investigated whether the downregulation of TSLP by Rh2 could affect the inhibition of Th2 responses in the DNCB-induced AD mouse model. Topical applications of DNCB led to an increase in the total number of CD4+ T-cells in the mouse tissues; however, administration of Rh2 was shown to result in decreased numbers of CD4+ T-cells. Furthermore, skin tissues treated with Rh2 exhibited significantly lower frequencies of IL-4-producing cells (Figure 6A). Consistent with the results in Figure 6a, Rh2 downregulated the mRNA expression of *IL-4* in the AD-induced NC/Nga mouse skin tissues (Figure 6B). These results collectively suggested that Rh2 can potently suppress the differentiation of IL-4-producing Th cells in AD pathogenesis.

Based on these results, it is hard to distinguish whether the inhibition of Th2 responses is regulated through only the down-regulated expression of TSLP in keratinocytes by Rh2 or through the direct inhibition of Th2 differentiation by Rh2. To investigate the possibility that Rh2 functions by directly inhibiting T-cell differentiation to Th2 cells, we evaluated the mRNA expression levels of *GATA3*, a transcription factor that plays a critical role in promoting CD4+ T-cell differentiation to Th2 cells and their function [32]. We also performed a fluorescence-activated cell sorting (FACS) analysis of the intracellular levels of IL-4 in differentiated Th2 cells. Accordingly, Rh2 significantly inhibited the expression of *GATA3* compared with Th2 polarized control (Figure 6C). Concomitantly, Rh2 decreased the frequency of IL-4 producing Th2 cells, suggesting that Rh2 could directly inhibit Th2 differentiation (Figure 6D). In addition, the concentration of IL-4 was significantly reduced following treatment with Rh2 (Figure 6E). These results suggested that Rh2 can directly affect Th2 cell differentiation as well as the effector functions of differentiated Th2 cells by regulating the expression of *GATA3*. 

## 3. Discussion

In this study, our observations provided evidence that Rh2 can effectively ameliorate AD-like skin symptoms via the inhibition of the expression of *TSLP* in keratinocytes following the suppression of the NF-κB signaling pathway, along with the modulation of Th2 cell differentiation and effector functions. 

TSLP acts as a master switch that induces the initiation and the development of AD and the atopic march [31]. Higher expression of TSLP in keratinocytes aggravates itch-evoked scratching that develops Th2 responses [12]. Thus, the regulation of TSLP in keratinocytes is thought to be important to prevent the development of severe allergic reactions [33,34]. Therefore, we identified the ginsenosides with the greatest inhibitory effects against TSLP expression in keratinocytes. While the anti-AD effects of ginsenosides had been previously suggested, the effects on the expression of TSLP and the identification of highly efficient ginsenosides had not been sufficiently investigated. 

Our results demonstrated that, of the 17 kinds of ginsenosides, Rh2 had the most potent anti-inflammatory activities by decreasing the expression of both TSLP and IL-8 in the stimulated NHKs. This study has certain strengths compared with previous research [16,17,35,36] in that we examined 17 different single ginsenosides to screen for the most effective ginsenoside at ameliorating AD. We hypothesized that the regulatory effects of ginseng extract and ginsenosides mixtures on alleviating AD may be mainly derived from the effects of Rh2.

Rh2 alleviated DNCB-induced AD-like symptoms in NC/Nga mice more potently compared with the DEX-treated mice, resulting in the downregulation of TSLP in the mouse skin tissues. While topical corticosteroids, including DEX, have reported side effects, including growth retardation and systemic actions, Rh2 has no harmful effects on body weight, indicating that growth retardation is not induced by Rh2. Furthermore, Rh2 did not induce splenomegaly, one of the characteristics of systemic immune dysfunction, suggesting that treatment with Rh2 is safe and subsequently that it could be an efficient agent for the treatment of AD.

With regard to the underlying mechanisms of Rh2, our results showed that Rh2 relieves AD by regulating TSLP expression in NHKs following the inhibition of the NF-κB signaling pathway, which is the main expression pathway for TSLP [37]. Our additional experiments demonstrated that Rh2 could regulate not only the expression of TSLP in keratinocytes but the differentiation of naïve T cells into Th2 cells directly. Although our study suggests regulatory effects of Rh2 on the expression of TSLP in keratinocytes and Th2 differentiation, additional experiments to identify the signaling mechanisms by transcriptomic or proteomic analysis and the effects of Rh2 on the TSLP–dendritic cells –Th2 axis [38] are required to improve our understanding and will be addressed in future studies. Though several questions still remain, this study is of importance, as we suggest that treatment with Rh2 alleviates AD by acting together to regulate TSLP in keratinocytes and to directly inhibit Th2 cell differentiation.

## 4. Materials and Methods

### 4.1. Cell Culture and Reagents

NHKs were purchased from American Type Culture Collection (ATCC, Manassas, VA, USA) and cultured in Dermal Cell Basal Medium (ATCC) supplemented with the Keratinocyte Growth kit (ATCC) at 37 °C with 5 % CO_2_. Ginsenosides compound K (C-K), F1, F2, gypenoside XVII (G17), gypenoside LXXV (G75), protopanaxadiol (PPD), protopanaxatriol (PPT), Rb1, Rb3, Rc, Rd, Re, Rg1, Rg2, Rg3, Rh1, and Rh2 were prepared as previously described [39,40]. PPD and PPT were purchased from Hongjiu Biotech Co. Ltd. (Dalian, China) and Da Nature Biological Engineering Co. Ltd. (Fusong, China), respectively. Ginsenoside C-K, F1, F2, G17, G75, Rg2, Rg3, Rh1, and Rh2 were transformed from PPD or PPT type ginsenosides using enzymatic bioconversion methods. Ginsenoside Rb1, Rb3, Rc, Rd, Re, and Rg1 were directly purified from PPD or PPT type ginsenosides using a silica column (Biotage, Uppsala, Sweden), an octadecyl silica (ODS) column (Biotage), and recycling preparative high-performance liquid chromatography (HPLC). The purity of each ginsenoside was greater than 95%. Ginsenosides were dissolved in dimethylsulfoxide (DMSO). DMSO, DEX, recombinant human TNF-α, Poly I:C, and bay 11-7082 were purchased from Sigma Aldrich (St. Louis, MO, USA). All kits were used as per the manufacturers’ instructions, unless otherwise specified.

### 4.2. Cell Viability Assay

Cell viability was determined using the MTT assay reagent (Promega, Madison, WI, USA). Briefly, 100 μL containing 5 × 10^4^ cells/mL of NHKs were seeded on a 96-well plate. Culture medium was replaced with fresh media containing various concentration of Rh2 (0, 1, 2.5, 5, and 10 μM) or DMSO [vehicle control, 0.05% (v/v)]. Cells were cultured for 24 h. Subsequently, 15 μL of the dye solution was applied to each well, and cells were further incubated. After 4 h, 100 μL of the solubilization solution/stop mix was added to each well. The absorbance at 570 nm was measured using a plate reader (Tecan, Männedorf, Switzerland). Cell viability was calculated as the ratio of the absorbance in Rh2-treated samples to that of the DMSO control.

### 4.3. Quantitative Real-Time Reverse Transcription PCR

Total RNA from the NHKs or skin tissues was isolated using the RNeasy mini kit (Qiagen, Hilden, Germany), analyzed by qRT-PCR using the Quantitect SYBR Green RT-PCR (Qiagen) on a CFX96 system (Bio-Rad, Hercules, CA, USA), and normalized with GAPDH.

### 4.4. Proteome Profiler Array

NHKs were treated with Rh2 (5 μM), or DMSO (vehicle control) and costimulated with 20 ng/mL TNF-α and 100 μg/mL Poly I:C for 24 h. After incubation, cell supernatants were collected and centrifuged for 20 min at 10,000 ×*g* to remove remaining cells. The inflammatory cytokines in the cell supernatants were assessed using a Proteome Profiler Human Cytokine Array Kit (R&D systems, Minneapolis, MN, USA). Densitometry of the dots was detected using ChemiDoc (Bio-Rad) and analyzed with Image J software (NIH, Bethesda, ME, USA).

### 4.5. Western Blot Analysis

The homogenates of murine dorsal skin tissues or collected NHKs were lysed with phenylmethylsulfonyl fluoride buffer containing phosphatase and protease inhibitors. The concentrations of the proteins were measured using a protein assay kit (Bio-Rad) and bovine serum albumin (BSA) as a standard. Proteins were separated using SDS-PAGE and transferred to polyvinylidene difluoride membranes (Invitrogen, Carlsbad, CA, USA). After blocking, membranes were incubated with anti- TSLP, p65, p-p65, p-IκBα, IκBα rabbit monoclonal antibodies (1:1000 dilution; Cell Signaling Technology, Danvers, MA, USA), and anti-β-actin mouse monoclonal antibody (1:1000 dilution; Santa Cruz Biotechnology, Dallas, TX, USA) for 2 h. Membranes were washed and then incubated with horseradish peroxidase-linked anti-rabbit or anti-mouse IgG secondary antibody (1:1000 dilution; Cell Signaling Technology) for 2 h. Immunoreactive proteins were detected using the enhanced chemiluminescence (ECL) solution (Pierce Biotechnology, Waltham, MA, USA). The densitometry of the bands was calculated using Image J (NIH).

### 4.6. Animal Models and Evaluation of AD Index

Male NC/Nga mice (7 weeks old) were purchased from ORIENTBIO (Sungnam, Korea). Mice were acclimated for one week and maintained at constant temperature (22 ± 3 °C) and humidity (30–70%) under a 12 h light-12 h dark cycle with free access to water and food. Animal experiments were conducted according to the guidelines of the Institutional Animal Care and Use Committee (IACUC) at Biotoxtech (No. 180355, Cheongju, Korea, 20 January 2015). AD was induced by topical application of DNCB (Sigma Aldrich), with minor modifications to the protocol [41,42]. Briefly, the dorsal skin of the mice was sensitized once in the first week by applying 200 μL of 1% DNCB dissolved in acetone:olive oil (3:1). Afterwards, 150 μL of 0.2% DNCB was topically applied on the same area twice per week for 3 weeks. The same area was treated with either 250 μL of phosphate buffered saline (PBS), DEX (10 μM), or Rh2 (5 μM) every day. The severity of the AD was evaluated once a week by scoring with an AD index, using previously described criteria [43]. Briefly, AD indices of 0 (none), 1 (mild), 2 (moderate), and 3 (severe) were evaluated according to 4 symptoms: (1) excoriation/erosion, (2) scarring/dryness, (3) erythema/hemorrhage, and (4) edema. The total sum of each score was considered as the obtained AD index.

### 4.7. Histological Analysis

Skin tissues were fixed in 4% paraformaldehyde right after excision from the mice. The fixed tissues were embedded in paraffin, sectioned, and then either stained with hematoxylin and eosin or with toluidine blue. Sections were analyzed to measure the infiltration of mast cells and eosinophils in the dermis area [42] and the thickness of the epidermis. The average of the 5-high-powered field readings was used for statistical analysis.

### 4.8. Immunohistochemistry and Immunofluorescence

Immunostaining was as previously described [44] with minor modifications. Skin tissues obtained from AD lesional mice were subjected to fixation in 4% paraformaldehyde, sectioning, deparaffinization, and rehydration. For immunohistochemistry, tissue slides were incubated in 0.01 M citrate buffer (pH 6.0) at 90 °C for 80 min, followed by blocking with normal rabbit serum (1:75, abcam, Cambridge, UK) for 20 min. Subsequently, slides were incubated with anti-TSLP antibody (1:75; abcam) for 2 h. Sections were then incubated with biotinylated anti-rabbit IgE and avidin-biotin horse radish peroxidase (HRP) complex (abcam) for 2 h and stained with diaminobenzidine (abcam) for 20 min. For immunofluorescence, slides were incubated with anti-IL-4 and anti-CD4 antibodies (1:50 dilution, 5 μg/mL, respectively; eBioscience, San Diego, CA, USA) for 1 h and then incubated with Alexa 488 or Alexa 647 conjugated secondary antibodies (1:200 dilution; abcam) plus 4’,6-diamidino-2-phenylindole (DAPI) for 1 h. All sections were analyzed using a Nikon i2 U microscope and the Nikon NIS-elements software (Tokyo, Japan).

### 4.9. Immunocytochemistry

NHKs were fixed with 4% paraformaldehyde, permeabilized with methanol, and blocked using an Immunofluorescence application solution kit (Cell Signaling Technology). NHKs were incubated with p65 antibody (1:400 dilution; Cell Signaling Technology) for 2 h and consecutively incubated with Alexa 594 conjugated goat anti-rabbit IgG secondary antibody (1:400 dilution; Thermo Fisher Scientific, Waltham, MA, USA) for 1 h in the dark. NHKs were additionally stained and mounted using ProLong Gold Antifade Mountant with DAPI (Thermo Fisher Scientific, Waltham, MA, USA).

### 4.10. Luciferase Assay

NHKs were transfected for 24 h with an NF-κB luciferase reporter vector and a negative control vector provided from the NF-κB reporter kit (BPS Bioscience, San Diego, CA, USA). Transfected NHKs were treated with DMSO, DEX, or Rh2 and co-stimulated with TNF-α (20 ng/mL) and Poly I:C (100 μg/mL) for 6 h. Cells were harvested, lysed, and prepared before subjected to the dual luciferase reporter assay (Promega, Madison, WI, USA). The activity of luciferase was measured using a Luminometer (Tecan), and relative luciferase activity was calculated as the fold change over the unstimulated vehicle control.

### 4.11. T-cell Differentiation

Mice were sacrificed by CO_2_ asphyxiation. Naive CD4+ T-cells were isolated from lymph nodes and the spleens dissected from the healthy male NC/Nga mice (8 weeks old) using the Naïve CD4+ T-cell isolation kit (Miltenyi Biotec, Bergisch Gladbach, Germany). The purity of the naïve CD4+ T-cells was >95%. Naïve CD4+ T-cells were differentiated into Th2 cells in the absence or the presence of Rh2 or DEX for 6 d using the CellXVIvo mouse Th2 cell differentiation kit (R&D systems). On day 6 of the culture, Th2-polarized cells were treated with phorbol-myristate acetate and inomycin (PMA/I) (R&D systems) and brefeldin A (Thermo Fisher Scientific) for 4 h. Cells were fixed, permeabilized using a Cytofix/Cytoper kit (BD Biosciences, San Jose, CA, USA), and stained with allophycocyanin (APC) conjugated anti-IL-4, fluorescein isothiocyanate (FITC) conjugated anti-CD4, and V450 conjugated anti-CD45 antibodies (abcam). Fluorescence was measured by flow cytometry (FACS LSRFortessa, BD Biosciences).

### 4.12. ELISA

TSLP and IL-8 were measured using the TSLP ELISA kit (abcam) and IL-8 ELISA kit (Thermo Fisher Scientific). To detect serum IgE, serum samples were collected from whole blood. Total serum IgE levels were measured using a mouse IgE ELISA kit (Thermo Fisher Scientific). For the detection of IL-4, naïve CD4+ T-cells were differentiated into Th2 cells for 6 d using the CellXVIvo mouse Th2 cell differentiation kit (R&D systems). IL-4 in the supernatant was measured by ELISA (abcam).

### 4.13. Statistical Analyses

Statistical significance between groups was determined by two-way analysis of variance using GraphPad PRISM (GraphPad Software, San Diego, CA, USA). Statistical differences were considered significant when (*) *p* < 0.05. All experiments were conducted independently at least 3 times, and results are expressed as mean ± SEM.

## 5. Conclusions

In summary, our results indicated that Rh2 was able to ameliorate AD symptoms by inhibiting the production of TSLP in keratinocytes via inhibition of the NF-κB signaling pathway as well as by modulating Th2 cell differentiation and their effector function, possibly through the regulation of the expression of *GATA3*. Taken together, our results implied that Rh2 could potentially be applied as an effective therapeutic agent in the clinical management of AD. 

## Figures and Tables

**Figure 1 ijms-20-06111-f001:**
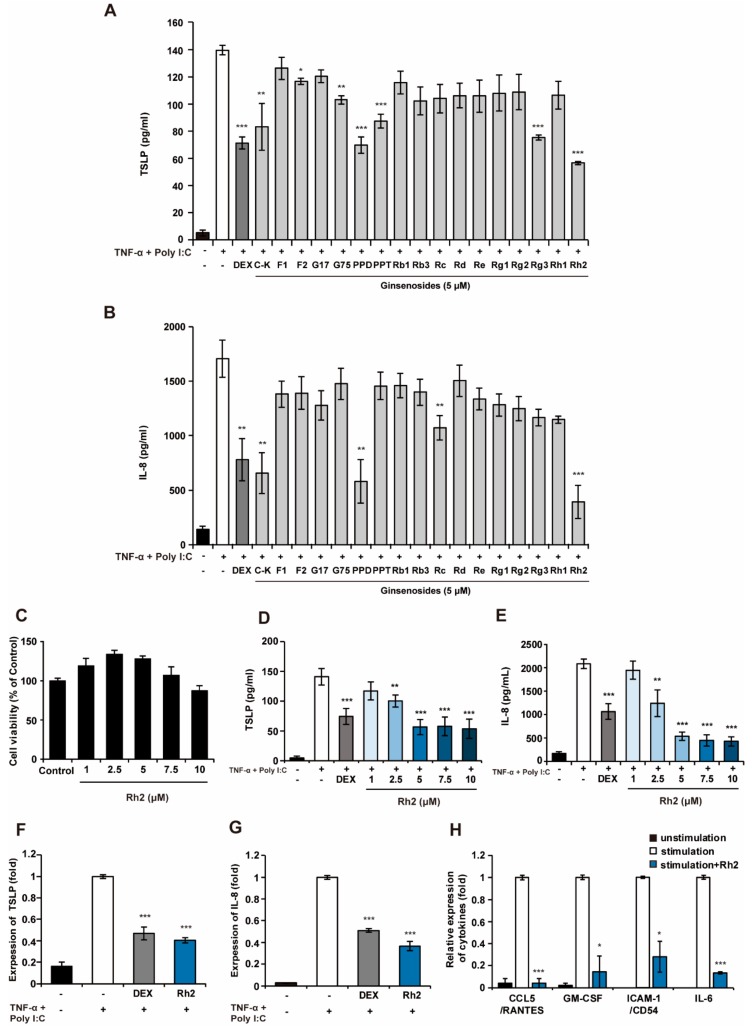
Rh2 significantly inhibits inflammatory cytokines in stimulated normal human keratinocytes (NHKs). NHKs were treated with dimethylsulfoxide (DMSO) (vehicle control), dexamethasone (DEX) (10 μM, positive control), ginsenosides (5 μM) (**A,B**), or increasing concentrations of Rh2 (**D,E**), and co-stimulated with TNF-α and Poly I:C for 24 h. The levels of thymic stromal lymphopoietin (TSLP) (**A,D**) and interleukin 8 (IL-8) (**B,E**) in the culture supernatants were measured by ELISA. (**C**) The viability of the NHKs was examined following treatments with DMSO [vehicle control, 0.05% (v/v)] or increasing concentrations of Rh2 for 24 h. (**F,G**) NHKs were treated with DMSO (vehicle control), DEX (10 μM, positive control), or Rh2 (5 μM) and co-stimulated with TNF-α and Poly I:C for 3 h. mRNA expression levels of *TSLP* (**F**) and *IL-8* (**G**) were analyzed by qRT-PCR, and normalized to glyceraldehyde 3-phosphate dehydrogenase (*GAPDH*). (**H**) NHKs were treated with DMSO or Rh2 (5 μM) and co-stimulated with TNF-α and Poly I:C for 24 h. Inflammatory cytokines in the cell supernatant were assessed using the proteome profiler array. * *p* < 0.05, ** *p* < 0.01, *** *p* < 0.001 vs the stimulated control.

**Figure 2 ijms-20-06111-f002:**
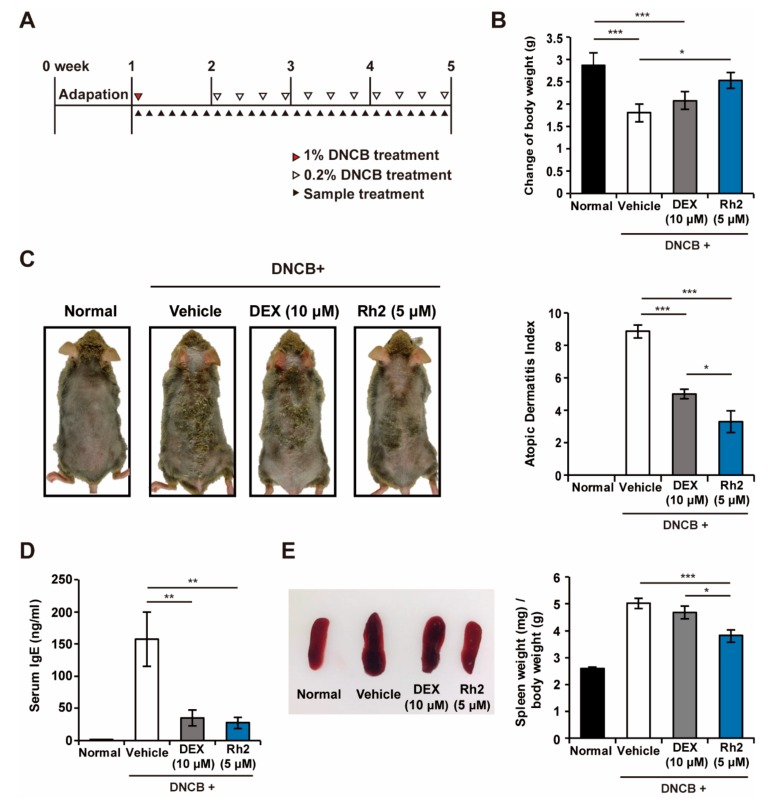
Rh2 attenuates atopic dermatitis (AD) symptoms in NC/Nga mice with 2,4-dinitrochlorobenzene (DNCB)-induced AD skin inflammation. (**A**) AD-like skin inflammation was induced in NC/Nga mice by topical applications of DNCB. DEX (10 μM, positive control) or Rh2 (5 μM) was topically applied on the dorsal skin of the mice as shown in the scheme. (**B**) Body weights were scored 28 days after AD induction. (**C**) Representative images showing the control and the AD-induced NC/Nga mice treated with DMSO (vehicle control), DEX (10 μM, positive control), or Rh2 (5 μM). AD index was evaluated by macroscopic observations. (**D**) Serum levels of IgE were measured by ELISA. (**E**) Representative images of the spleens excised from control and treated mice. Spleen weight was measured at the end of the experiment. * *p* < 0.05, ** *p* < 0.01, *** *p* < 0.001. *n* = 6.

**Figure 3 ijms-20-06111-f003:**
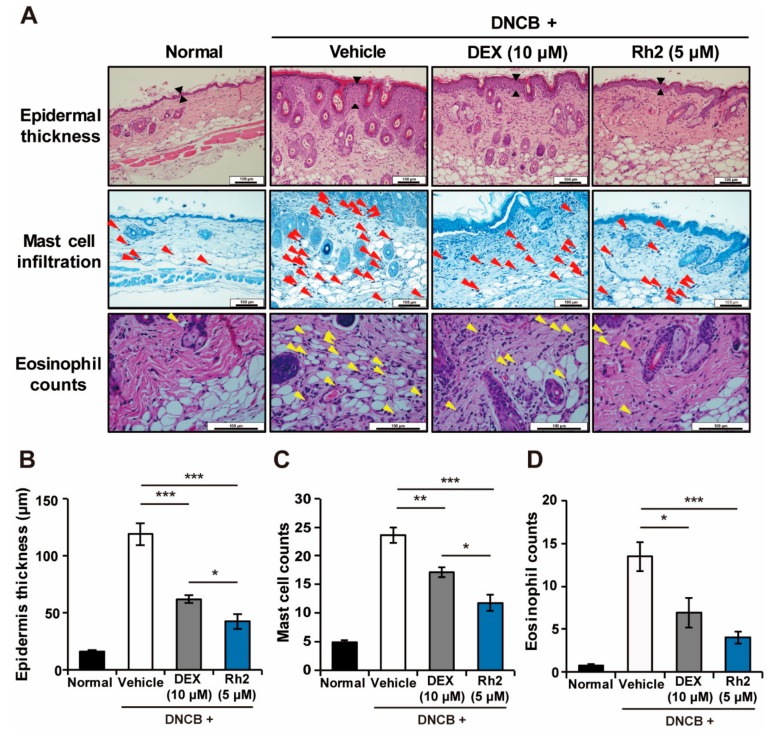
Inhibitory effects of Rh2 on AD symptoms in skin tissues of NC/Nga mice. (**A**) Representative histological images of the dorsal skin lesions in normal mice and mice with DNCB-induced AD topically treated with DMSO (vehicle control), DEX (10 μM, positive control), or Rh2 (5 μM). Red and yellow arrowheads indicate mast cells and eosinophils, respectively. (**B**) The thickness of the epidermis of the back skin was measured. (**C**,**D**) The number of infiltrated mast cells or eosinophils was counted in five representative high-power fields. Scale bar = 100 μm. * *p* < 0.05, ** *p* < 0.01, *** *p* < 0.001. *n* = 6.

**Figure 4 ijms-20-06111-f004:**
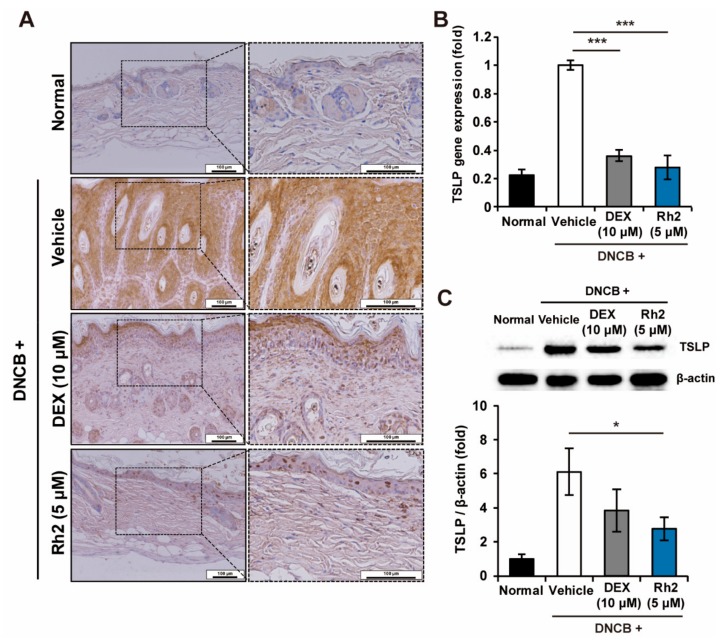
Topical applications of Rh2 suppressed mRNA and protein expression of TSLP in NC/Nga mice with DNCB-induced AD. (**A**) Immunohistochemical evaluation of the inhibitory effects of Rh2 on the production of TSLP in the skin tissues of mice. Scale bar = 100 μm. (**B**) The relative expression of *TSLP* was analyzed by qRT-PCR in the skin tissues from the AD mice models, normalized to *GAPDH*. (**C**) Immunoblotting analysis of the protein expression level of TSLP in the skin tissues of the AD mice. The densitometry of the bands was calculated using Image J software. * *p* < 0.05, *** *p* < 0.001. *n* = 6.

**Figure 5 ijms-20-06111-f005:**
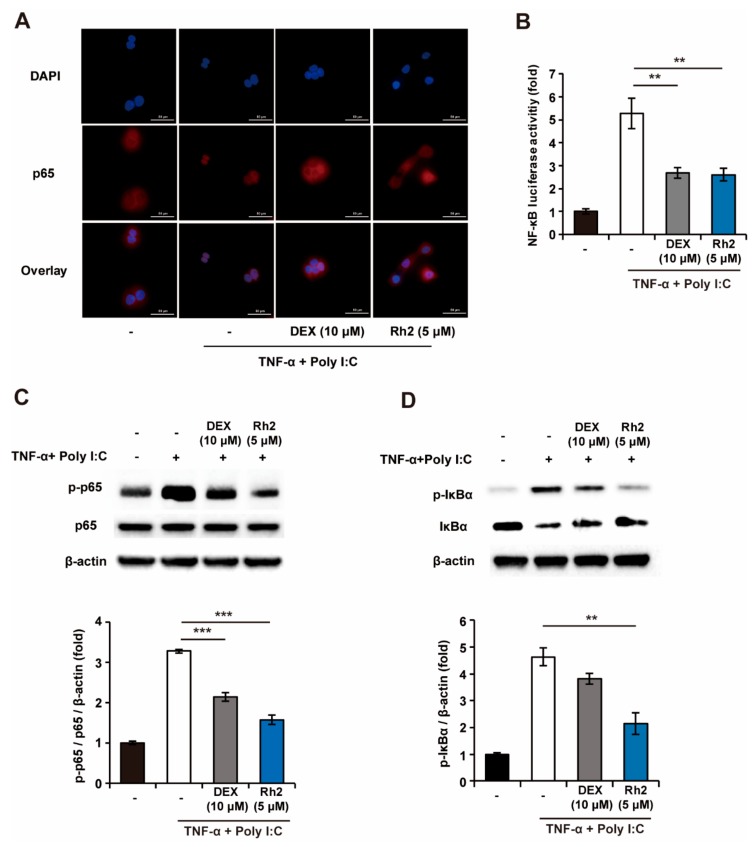
Rh2 significantly inhibits the NF-κB signaling pathway in NHKs. (**A**) NHKs were treated with DMSO (vehicle control), DEX (10 μM, positive control), Rh2 (5 μM), and sensitized with TNF-α (20 ng/mL) and Poly I:C (100 μg/mL) for 1 h. The suppressive effects of Rh2 on NF-κB (p65) nuclear translocation were analyzed by immunofluorescent imaging (scale bar = 50 μm). (**B**) The effects of the Rh2 on the NF-κB promoter activity were analyzed using the NF-κB luciferase assay. (**C**,**D**) NHKs were treated with DMSO (vehicle control), DEX (10 μM, positive control), or Rh2 (5 μM) and co-stimulated with TNF-α and Poly I:C for 1 h. The inhibitory effects of the Rh2 on the phosphorylated protein levels of the p65 and IκBα degradation were confirmed by immunoblotting analysis. ** *p* < 0.01, *** *p* < 0.001. *n* = 3.

**Figure 6 ijms-20-06111-f006:**
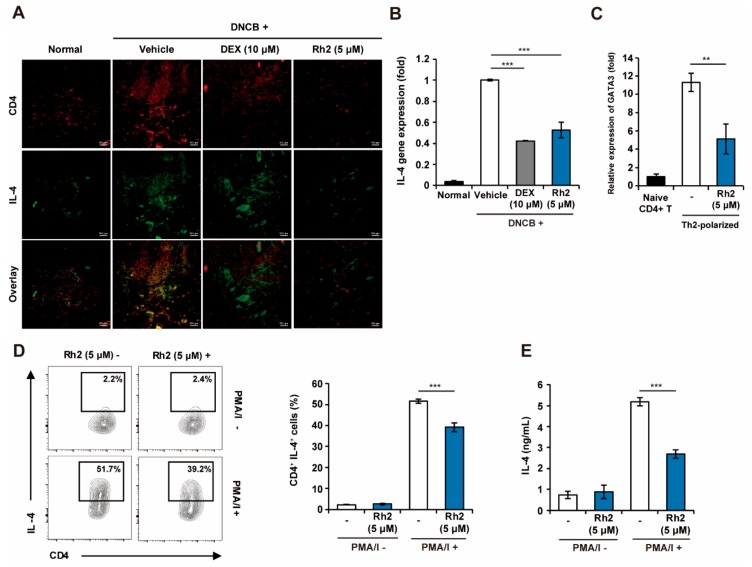
Rh2 inhibits the differentiation of CD4+ T-cells to Th2 cells. (**A**) CD4+ T-cells and IL-4 in skin tissues obtained from DNCB-induced AD mice were stained with allophycocyanin (APC) conjugated CD4+ antibody or fluorescein isothiocyanate (FITC) conjugated IL-4 antibody, respectively. (**B**) mRNA expression levels of *IL-4* in control and treated skin tissues were analyzed by qRT-PCR and normalized to *GAPDH*. (**C**,**D**) Naïve CD4+ T-cells from healthy NC/Nga mice were differentiated to Th2 cells by incubation in the Th2 conditioning media in the presence of DMSO (vehicle control) or Rh2 (5 μM). On day six of the culture, the Th2-polarized cells were treated with phorbol-myristate acetate and inomycin (PMA/I) and brefeldin A for 4 h and analyzed by qRT-PCR and fluorescence-activated cell sorting (FACS), respectively. (**E**) Naïve CD4+ T-cells were differentiated to Th2 cells during treatments with Rh2 for 6 d. Th2-polarized cells were pretreated with Rh2 (5 μM) for 30 min and stimulated with PMA/I for 24 h. Levels of IL-4 in the cell supernatants were measured by ELISA. ** *p* < 0.01, *** *p* < 0.001. *n* = 6 (A,B), *n* = 3 (C,D).

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
