# Peer review of "Ginsenoside Rh2 Ameliorates Atopic Dermatitis in NC/Nga Mice by Suppressing NF-kappaB-Mediated Thymic Stromal Lymphopoietin Expression and T Helper Type 2 Differentiation"

_ijms, 2019, doi:10.3390/ijms20246111_

Round 1

Reviewer 1 Report

The submitted manuscript (ijms-656298) shows the potentail activity of ginsenoside Rh2 in the atopic dermatitis treatment. The request for Authors is providing more detailed data on tested ginsenosides in the section of Material and methods L. 269-270. Please describe the way of compounds preparation in the manuscript. Were the compounds isolated or purchased?  In addition, what was the concentration of DMSO applied into the cell culture. Please verify it(L. 287, Figure 1 etc.). What about its cytotoxicity?

In addition, please provide the concentration of Rh2 in figures 2-6.

L. 74 Providing the names of compounds requires any introduction, e.g. in the Introduction section.

Minor comments:

L. 81 instead of „for”, should be „at”

Please change the words or statements according to native speaker advices L. 86 „delineate”, L. 157 „corroborate”, L. 171 „unveil”

L. 126 Please check „RH2”

L. 132 Please develop „28 d”

Author Response

Response to Reviewer 1 Comments

 Dear Reviewer 1,

We appreciate your insightful comments that are very helpful for us to further improve our manuscript. We have carefully revised the manuscript per your suggestion and comments. Their inputs have improved the quality of our manuscript, and the revised version presents a better and more balanced description of our manuscripts. Our responses to your questions and comments are detailed below.

Comments and Suggestions for Authors

Point 1: The submitted manuscript (ijms-656298) shows the potentail activity of ginsenoside Rh2 in the atopic dermatitis treatment. The request for Authors is providing more detailed data on tested ginsenosides in the section of Material and methods L. 269-270. Please describe the way of compounds preparation in the manuscript. Were the compounds isolated or purchased?  

Response 1: We thank the reviewer for these comments. According to the reviewer’s comment, original sentence was revised to state as “Ginsenosides compound K (C-K), F1, F2, gypenoside XVII (G17), gypenoside LXXV (G75), protopanaxadiol (PPD), protopanaxatriol (PPT), Rb1, Rb3, Rc, Rd, Re, Rg1, Rg2, Rg3, Rh1, and Rh2 were prepared as previously described. Briefly, PPD and PPT were purchased from Hongjiu Biotech Co. Ltd (Dalian, China) and Da Nature Biological Engineering Co. Ltd (Fusong, China), respectively. Ginsenoside C-K, F1, F2, G17, G75, Rg2, Rg3, Rh1, and Rh2 were transformed from PPD or PPT type ginsenosides using enzymatic bioconversion methods. Ginsenoside Rb1, Rb3, Rc, Rd, Re, a nd Rg1 were directly purified from PPD or PPT type ginsenosides using a silica column (Biotage, Uppsala, Sweden), an ODS column (Biotage), and recycling preparative HPLC. The purity of each ginsenoside was greater than 95%. Ginsenosides were dissolved in dimethylsulfoxide (DMSO).” on page 20, in lines 285-294 in the materials and methods of the revised manuscript.

Point 2: In addition, what was the concentration of DMSO applied into the cell culture.

Please verify it (L. 287, Figure 1 etc.). What about its cytotoxicity?

Response 2: We thank you very much for your comments. We added “0.05% (v/v))” on page 20, in line 301 in the materials and methods part of the revised manuscript. We additionally revised to “DMSO (vehicle control, 0.05% (v/v)) or” on page 5, in line 107 in the legend of figure 1.

  We confirmed that 0.05% of DMSO has not shown cytotoxicity on normal human keratinocytes. There was no difference of cell viability between non-treated control and DMSO-treated control. (data are not shown)

Point 3: In addition, please provide the concentration of Rh2 in figures 2-6.

Response 3: We agreed with the reviewer on this point. According to the reviewer’s comment, the concentration of Rh2 was added in figures 2-6.

Point 4: L. 74 Providing the names of compounds requires any introduction, e.g. in the Introduction section

Response 4: We thank the reviewer for these comments. To provide more details of ginsenosides, we added the chemical structure and full name of all tested ginsenosides in supplementary figure 1.

Point 5: Minor comments:

81 instead of „for”, should be „at” -> at

Response 5: We thank you very much for your comments, and revised to “at” on page 3, in line 90 in the results section of the revised manuscript.

Point 6: Please change the words or statements according to native speaker advices

86 „delineate”, L. 157 „corroborate”, L. 171 „unveil” 126 Please check „RH2” 132 Please develop „28 d”

Response 6: We thank you for your comments. We revised “delineate” to “demonstrate” on page 3, in line 95, “corroborate” to “further verify” on page 11, in line 171, “unveil” to “elucidate” on page 13, in line 186, “RH2” to “Rh2” on page 6, in line 137, and “28 d” to “28 days” on page 8, in line 144.

Again, we thank the reviewer and editor very much for the valuable comments and helping us to significantly improve our manuscript. In summary, we have addressed each of the points raised by the reviewer in the revised manuscript. We hope that these revisions have addressed the reviewer’s comments, and that the revised manuscript will now be acceptable for publication in International journal of molecular sciences. If you have any questions, please do not hesitate to contact us.

Thank you very much.

Sincerely yours,

Prof. Sun Chang Kim

Reviewer 2 Report

Abstract section:

The use of acronyms should be improved and made uniform. Before using an acronym, the extende name has to be reported.

Introduction section:

The description of Ginsenosides is limited. Authors should provide a more detailed background including phytochemical data and pharmacological description of mechanism of action.

Figure 1A e 1B: These figures are not clear. They should be enlarged. Additionally, the acronyms are not described in the results section. Please, amend this aspect.

The description of real-time analysis paradigm is too synthetic. Please, amend the related paragraph.

The animal study description does not show the authorization number for the procedures. Please include the missing information.

Author Response

Response to Reviewer 2 Comments

Dear Reviewer 2,

We appreciate your insightful comments that are very helpful for us to further improve our manuscript. We have carefully revised the manuscript per your suggestion and comments. Their inputs have improved the quality of our manuscript, and the revised version presents a better and more balanced description of our manuscripts. Our responses to your questions and comments are detailed below.

Comments and Suggestions for Authors

Point 1: Abstract section:

The use of acronyms should be improved and made uniform. Before using an acronym, the extende name has to be reported.

Response 1: We thank the reviewer for these comments. To make uniformly, we added “ginsenoside” on page 1, in line 21, revised “IgE” to “immunoglobulin E” on page 1, in line 27, and deleted “Th2” on page 1, in line 31 in the abstract of the revised manuscript.

Point 2: Introduction section:

The description of Ginsenosides is limited. Authors should provide a more detailed background including phytochemical data and pharmacological description of mechanism of action.

Response 2: We thank you very much for your comments. According to your comments, we have added a detailed background of phytochemical data of ginsenosides by inserting “Ginsenosides are triterpene saponins that consist of a dammarane skeleton with a variety of sugar moieties attached to the C-3 and C-20 positions. The number, position, and type of sugar moieties have been known to contribute to diverse pharmacological potentials of ginsenosides, such as anti-cancer, anti-aging, and anti-inflammatory properties.” on page 3, in line 53-58 in the introduction part of the revised manuscript.

We additionally added the detailed description of pharmacological mechanism of action of ginseng and ginsenosides by inserting “via inhibition of mitogen-activated protein kinase (MAPK) and NF-κB pathway” on page 2, in line 60, and “by suppressing the expressions of cyclooxygenase (COX)-2, interleukin (IL)-1β, tumor necrosis factor-α (TNF-α) and interferon-γ (IFN-γ)” on page 2, in line 62-64 in the introduction of the revised manuscript.

Point 3: Figure 1A e 1B: These figures are not clear. They should be enlarged. Additionally, the acronyms are not described in the results section. Please, amend this aspect.

Response 3: We agree with the reviewer on this point. We have enlarged the figure 1A and 1B and added the description of the acronyms of ginsenosides on page 2, in line 76-78 in the results section.

Point 4: The description of real-time analysis paradigm is too synthetic. Please, amend the related paragraph.

Response 4: We thank the reviewer for these comments. To describe the results of real-time analysis more detail, we separated the explains about the results of TSLP and IL-8, respectively by revising “This effect was further confirmed by qRT-PCR analysis, resulting administration of 5 μM of Rh2 significantly downregulated the mRNA expression level of TSLP (Figure 1F). In addition, the mRNA expression level of IL-8 was also suppressed by treatment of Rh2 (Figure 1G).” on page 3, in line 92-95 in the results section.

In addition, we divided the description about the results of mRNA expression of TSLP and protein expression of TSLP by revising “The mRNA expression of TSLP was significantly downregulated by Rh2 treatment in the mouse skin tissues (Figure 4B). In accordance with Figure 4A-B, protein expression of TSLP was also significantly decreased in the mouse skin tissues subjected to topical applications of Rh2 (Figure 4C).” on page 11, in line 172-175 in the results part of the revised manuscript.

We additionally added “compared with Th2-polarized control” on page 17, in line 226-227 in the results section to explain more detail.

Point 5: The animal study description does not show the authorization number for the procedures. Please include the missing information.

Response 5: We thank you very much for your comments. The authorization number of the procedures of animal experiments has been added on page 21, in line 335-336 in the materials and methods of the revised manuscript.

Again, we thank the reviewer and editor very much for the valuable comments and helping us to significantly improve our manuscript. In summary, we have addressed each of the points raised by the reviewer in the revised manuscript. We hope that these revisions have addressed the reviewer’s comments, and that the revised manuscript will now be acceptable for publication in International journal of molecular sciences. If you have any questions, please do not hesitate to contact us.

Thank you very much.

Sincerely yours,

Prof. Sun Chang Kim